# Multivariate analysis reveals shared genetic architecture of brain morphology and human behavior

Ronald de Vlaming [1,10], Eric A. W. Slob[2,3,4,10], Philip R. Jansen [5,6], Alain Dagher [7], Philipp D. Koellinger[1,8], Patrick J. F. Groenen [9] & Cornelius A. Rietveld [2,3✉]

Human variation in brain morphology and behavior are related and highly heritable. Yet, it is largely unknown to what extent specific features of brain morphology and behavior are genetically related. Here, we introduce a computationally efficient approach for multivariate genomic-relatedness-based restricted maximum likelihood (MGREML) to estimate the genetic correlation between a large number of phenotypes simultaneously. Using individual-level data ($N = 20{,}190$) from the UK Biobank, we provide estimates of the heritability of gray-matter volume in 74 regions of interest (ROIs) in the brain and we map genetic correlations between these ROIs and health-relevant behavioral outcomes, including intelligence. We find four genetically distinct clusters in the brain that are aligned with standard anatomical subdivision in neuroscience. Behavioral traits have distinct genetic correlations with brain morphology which suggests trait-specific relevance of ROIs. These empirical results illustrate how MGREML can be used to estimate internally consistent and high-dimensional genetic correlation matrices in large datasets.

[1] School of Business and Economics, Vrije Universiteit Amsterdam, Amsterdam, The Netherlands. [2] Department of Applied Economics, Erasmus School of Economics, Rotterdam, The Netherlands. [3] Erasmus University Rotterdam Institute for Behavior and Biology, Erasmus School of Economics, Rotterdam, The Netherlands. [4] MRC Biostatistics Unit, School of Clinical Medicine, University of Cambridge, Cambridge, UK. [5] Department of Complex Trait Genetics, Center for Neurogenomics and Cognitive Research, Amsterdam Neuroscience, Vrije Universiteit Amsterdam, Amsterdam, The Netherlands. [6] Department of Clinical Genetics, VU Medical Center, Amsterdam UMC, Amsterdam, The Netherlands. [7] Montreal Neurological Institute, McGill University, Montreal, Quebec, Canada. [8] La Follette School of Public Affairs, University of Wisconsin-Madison, Madison, WI, USA. [9] Econometric Institute, Erasmus School of Economics, Rotterdam, The Netherlands. [10] These authors contributed equally Ronald de Vlaming, Eric A.W. Slob. ✉email: nrietveld@ese.eur.nl

Global and regional gray matter volumes are known to be linked to differences in human behavior and mental health[1]. For example, reduced gray matter density has been implicated in a wide range of neurodegenerative diseases and mental illnesses[2–5]. In addition, differences in gray matter volume have been related to cognitive and behavioral phenotypic traits such as fluid intelligence and personality, although results have not always been replicable[6,7].

Variation in brain morphology can be measured noninvasively using magnetic resonance imaging (MRI). Large-scale data collection efforts, such as the UK Biobank[8], that include both MRI scans and genetic data have enabled recent studies to discover the genetic architecture of human variation in brain morphology and to explore the genetic correlations of brain morphology with behavior and health[9–13]. These studies have demonstrated that all features of brain morphology are genetically highly complex traits and that their heritable component is mostly due to the combined influence of many common genetic variants, each with a small effect.

A corollary of this insight is that even the currently largest possible genome-wide association studies (GWASs) were only able to identify a small portion of the genetic variants underlying the heritable components of brain morphology: The vast majority of their heritability remains missing[9–14]. As a consequence, the genetic correlations of regional brain volumes with each other, as well as with human behavior and health have remained largely elusive. However, such estimates could advance our understanding of the genetic architecture of the brain, for example, regarding its structure and plasticity. Similarly, a strong genetic overlap of specific features of brain morphology with mental health would provide clues about the neural mechanisms behind the genesis of disease[15–17].

We developed multivariate genomic-relatedness-based restricted maximum likelihood (MGREML) to provide a comprehensive map of the genetic architecture of brain morphology. MGREML overcomes several limitations of existing approaches to estimate heritability and genetic correlations from molecular genetic (individual-level) data. Contrary to existing pairwise bivariate approaches, MGREML guarantees internally consistent (i.e., at least positive semidefinite) genetic correlation matrices and it yields standard errors that correctly reflect the multivariate structure of the data. The software implementation of MGREML is computationally substantially more efficient than both the traditional bivariate genomic-relatedness-based restricted maximum likelihood (GREML)[18,19] and comparable multivariate approaches[20–24]. Moreover, we show that MGREML allows for stronger statistical inference than methods that are based on GWAS summary statistics, such as bivariate linkage-disequilibrium (LD) score regression (LDSC)[25,26]. In short, MGREML yields precise and internally consistent estimates of genetic correlations across a large number of traits when existing approaches applied to the same data are either less precise or computationally unfeasible.

We leverage the advantages of MGREML by analyzing brain morphology based on MRI-derived gray matter volumes in 74 regions of interest (ROIs). We also estimate the genetic correlations of these ROIs with global measures of brain volume and eight human behavioral traits that have well-known associations with mental and physical health. The anthropometric measures height and body-mass index are also analyzed, because of their relationships with brain size[6,13]. Our analyses are based on data from the UK Biobank brain imaging study[27].

## Results
### Estimating genetic correlations.
Several methods can be used to estimate heritabilities and genetic correlations from molecular genetic data on single-nucleotide polymorphisms (SNPs). One class of these methods is based on GWAS summary statistics[25,26,28]. Another class of methods is based on individual-level data, such as GREML and variations of this approach[22–24,29–33]. Methods based on GWAS summary statistics such as LDSC[25,26] and variants thereof[34] can leverage the ever-increasing sample sizes of GWAS meta- or mega-analyses[35]. These methods are computationally efficient and benefit from the fact that GWAS summary statistics are often publicly shared[36,37]. However, the computationally more intensive methods based on individual-level data, such as GREML are statistically more powerful[38]. That is, the resulting estimates are more precise as reflected in the size of the standard errors.

Due to the high costs of MRI brain scans, GWAS meta-analysis samples for brain imaging genetics are still relatively small compared to GWAS meta-analysis samples for traits that can be measured at low cost (e.g., height[39] and educational attainment[40]). The UK Biobank brain imaging study (Methods) is currently by far the largest available sample that includes both MRI scans and genetic data, often surpassing the sample size of most previous studies in neuroscience by an order of magnitude or more[9,10,13]. Therefore, this dataset is particularly suitable for our individual-level data analysis.

Irrespective of whether one uses GWAS summary statistics or individual-level data, the use of bivariate methods poses another challenge when computing genetic correlation across more than two traits. In this case, the correlation estimates from bivariate analyses of all pairwise combinations of traits are often simply stacked, to form a 'grand' correlation matrix[25,26,41]. However, this 'pairwise bivariate' approach can result in genetic correlation matrices that are not internally consistent (i.e., they describe interrelationships across traits that cannot exist simultaneously). In mathematical terms, the resulting matrices can be indefinite. Although the correlation between two traits can vary between $-1$ and $+1$, their correlations with a third trait are naturally bounded. For a set of three traits, the solution is positive (semi)-definite when the correlations satisfy the following condition: $r_{12}^2 + r_{13}^2 + r_{23}^2 - 2r_{12}r_{13}r_{23} \le 1$, where $r_{st}$ denotes the correlation between traits $s$ and $t$. This condition is violated, for instance, when pairwise correlations are estimated to be $r_{12} = 0.9$, $r_{13} = 0.9$, and $r_{23} = 0.2$. In fact, the genetic correlation matrix in the well-known atlas of genetic correlations is not positive semidefinite[25]. A second consequence of the pairwise bivariate approach is that the standard errors of the resulting genetic correlation matrix do not adequately reflect the multivariate structure of the data.

**MGREML.** Our multivariate extension of GREML estimation[18,32] guarantees the internal consistency of the estimated genetic correlation matrix by adopting an appropriate factor model for the variance matrices (Supplementary Note 1). An important benefit of this approach is that estimates are always valid, in the sense that the likelihood is defined, even within the optimization procedure. Joint estimation also ensures that the standard errors of the estimated genetic correlations reflect the multivariate structure of the data correctly. Therefore, methods such as genomic structural equation modelling (genomic SEM)[42] that use multivariate genetic correlation matrices as input may benefit from using MGREML results, by avoiding the potentially distorting pre-processing step of bending[43] an indefinite genetic correlation matrix. To deal with the computational burden and to make MGREML applicable to large data sets in terms of individuals and traits, we derived efficient expressions for the likelihood function and developed a rapid optimization algorithm (Supplementary Note 1). In Supplementary Note 3, we show that MGREML is

computationally faster than pairwise bivariate GREML. Moreover, comparisons with ASReml[20], BOLT-REML[23], GEMMA[22], MTG2[24], and WOMBAT[21] highlight the computational gains afforded by MGREML. That is, none of these software packages is able to deal with the dimensionality of our empirical application. Finally, a comparison of results obtained with MGREML with results obtained using LDSC shows that standard errors obtained with MGREML are 32.7–50.6% smaller, illustrating the substantial gains in statistical power afforded by MGREML.

**Analysis of brain morphology**. We used MGREML to analyze the heritability of and genetic correlations across 86 traits in 20,190 unrelated 'white British' individuals from the UK Biobank (Fig. 1, Methods). The subset of 76 brain morphology traits includes total brain volume (gray and white matter), total gray matter volume, and gray matter volumes in 74 regions of interest (ROIs) in the brain. Relative volumes were obtained by dividing ROI gray matter volumes by total gray matter volume. The full set of heritability estimates is available in Supplementary Data 1. Figure 2a, b show that SNP-based heritability ($h^2_{\mathrm{SNPs}}$) (i.e., the proportion of phenotypic variance which can be explained by autosomal SNPs) is on average highest in the insula, and in the cerebellar and subcortical structures of the brain (average $h^2_{\mathrm{SNPs}}$ is 33.1, 32.4, and 29.5%, respectively, with corresponding standard errors of 0.019 for all) and lowest in the parietal, frontal, and temporal lobes of the cortex (average $h^2_{\mathrm{SNPs}}$ is 21.2, 21.4, and 25.2%, respectively, with corresponding standard errors of 0.019 for all). Grouping of the $h^2_{\mathrm{SNPs}}$ estimates in networks of intrinsic functional connectivity[44] reveals that ROIs in the heteromodal cortex (frontoparietal, dorsal attention) are less heritable than primary (visual, somatomotor), subcortical and cerebellar regions (Fig. 3a).

The full set of estimated genetic correlations ($r_g$) is available in Supplementary Data 1. Using spatial mapping, Fig. 2c visualizes the estimated genetic correlations across the relative volumes of the cortical and subcortical brain areas. The largest positive genetic correlations were found between the insular and frontal regions (average $r_g = 0.17$) and between the cerebellar and subcortical areas (average $r_g = 0.15$). The largest negative correlations were present between the cerebellar and insular regions (average $r_g = -0.18$) and between the cerebellar and frontal regions (average $r_g = -0.15$) (Fig. 2d). Figure 3b shows that the genetic correlations are particularly strong within intrinsic connectivity networks, especially the visual, somatomotor, subcortical, and cerebellum networks, possibly because of lower experience-dependent plasticity in these brain regions compared to heteromodal and associative areas[45]. Using Ward's method for hierarchical clustering[46], we identify four clusters

within the estimated genetic correlations for the 74 ROIs in the brain (Fig. 4). The first cluster (18 ROIs) includes most of the frontal cortical areas of the brain, the second (18 ROIs) the cerebellar cortex, the third (18 ROIs) subcortical structures including the brain stem, and the last cluster (20 ROIs) contains a mixture of temporal and occipital brain areas.

We also used MGREML to estimate the genetic correlations between brain morphology and eight human behavioral traits that are known to be related to health and that have previously been studied in large-scale GWASs, as well as the anthropometric measures height and body-mass index. Statistically significant correlations are highlighted in Supplementary Data 1 (Panel c). Spatial maps of the genetic correlation between brain morphology and the behavioral traits are shown in Fig. 5. For subjective well-being, we find the strongest genetic correlation with the Middle Frontal Gyrus (Fig. 5a, $r_g = 0.21$, corresponding standard error 0.088), a region that has been linked before to emotion regulation[47]. The genetic correlations of the ROIs with neuroticism (Fig. 5b) and depression (Fig. 5c) are generally weak and insignificant, potentially reflecting the coarseness of these phenotypic measures in the UK Biobank data. The strongest genetic correlation with the number of alcoholic drinks consumed per week is with the Lateral Occipital Cortex, superior and inferior divisions (Fig. 5d, $r_g = 0.23$ and $r_g = 0.18$, respectively, corresponding standard errors 0.106 and 0.092). Although the phenotypic correlations between the analyzed ROIs and alcohol consumption are generally negative[48], these particular brain regions are among those implicated in the affective response to drug cues based on the perception-valuation-action model[49]. For educational attainment and intelligence, the strongest correlations are found in the frontal lobe region ($r_g = -0.13$, corresponding standard error 0.065, between educational attainment and the Superior Frontal Gyrus, and $r_g = 0.16$, corresponding standard error 0.056, between intelligence and the Frontal Medial Cortex). Figure 5e, f show that the genetic correlation structures estimated for educational attainment and intelligence are largely similar, in line with earlier studies showing the strong genetic overlap between these two traits[50]. Genetic correlations of the ROIs with visual memory (Fig. 5g) are insignificant, and the strongest genetic correlation of reaction time is with the Middle Temporal Gyrus, temporooccipital part (Fig. 5h, $r_g = 0.20$, corresponding standard error 0.085). Activity within the middle temporal gyrus has been linked before with reaction time[51].

Earlier studies suggest that the size of the brain is positively associated with traits such as intelligence[6]. When analyzing absolute brain volumes of the ROIs rather than relative brain volumes (i.e., relative to total gray matter volume in the brain), we indeed observe robust positive relationships between the absolute

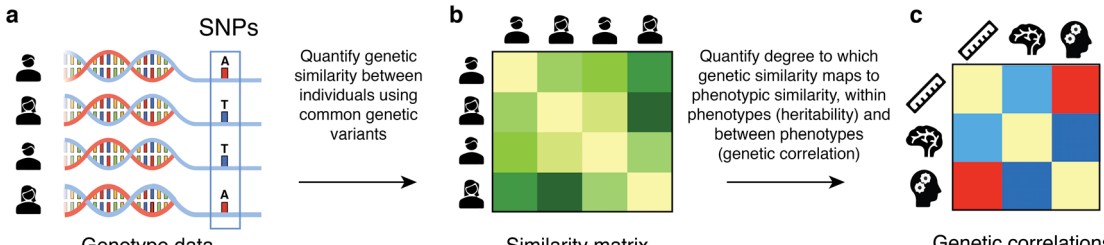

**Fig. 1 Visualization of multivariate genomic-relatedness-based restricted maximum likelihood (MGREML). a** Common genetic variants (single-nucleotide polymorphisms, "SNPs") in the human genome **b** are used to construct a genomic-relatedness matrix (GRM) capturing pairwise genetic similarity between individuals in the sample. **c** MGREML uses this GRM to jointly estimate heritabilities of phenotypes and genetic correlations ($r_g$) across multiple phenotypes, by quantifying the degree to which genetic similarity maps to phenotypic similarity (across all individuals and phenotypes in the sample). In our empirical application, 1,384,830 common SNPs are used to analyze the genetic correlations across $T = 86$ phenotypes in a sample of $N = 20,190$ unrelated individuals.

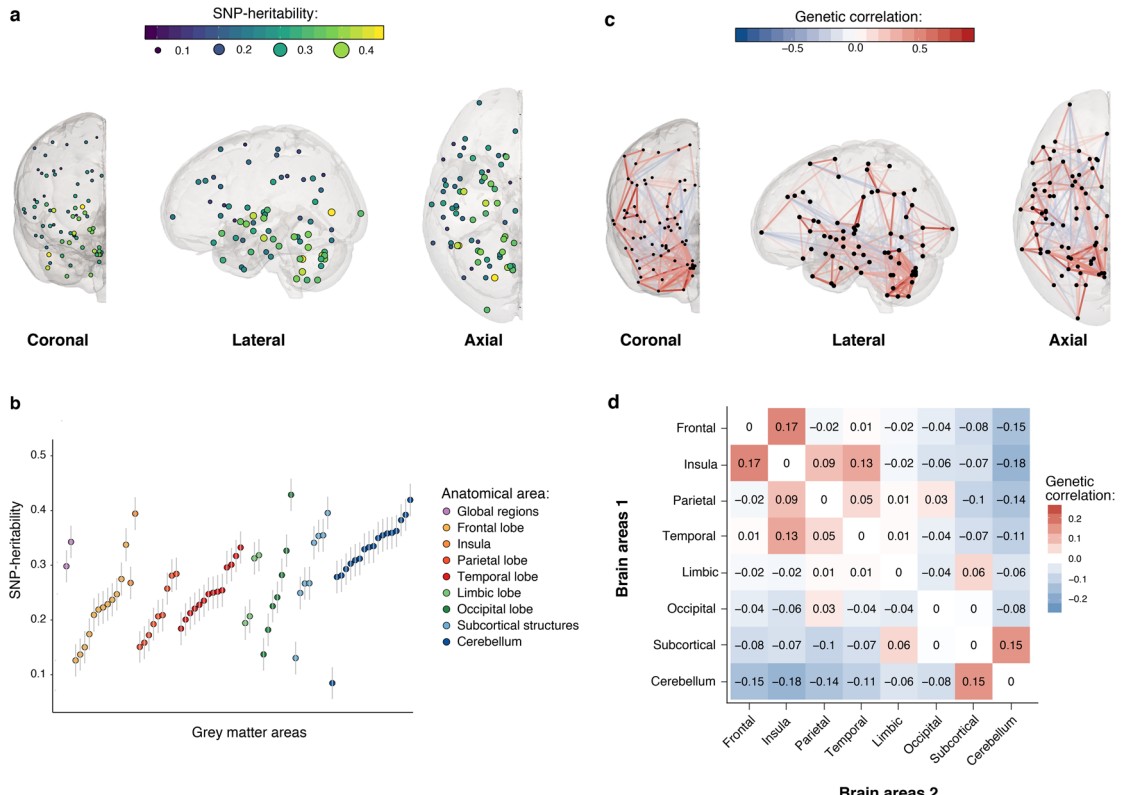

**Fig. 2 Spatial mapping of SNP-based heritability and genetic correlation estimates obtained using MGREML ($N = 20{,}190$) of relative gray matter volumes in different cortical and subcortical brain areas. a** SNP-based heritability of relative gray matter volume mapped to the respective brain region in three dimensions. Each dot represents an area, the color and size represent the heritability of that area. **b** SNP-based heritability and standard error of relative gray matter volume of each brain region grouped by global anatomical area. **c** Genetic correlations between the cortical and subcortical relative gray matter volumes. The opacity and color represent the strength of the genetic overlap between these two areas (blue vertices represent a negative correlation, red vertices a positive correlation). Only genetic correlations larger than |0.25| are shown. **d** Average genetic correlations in broad anatomical areas of the brain.

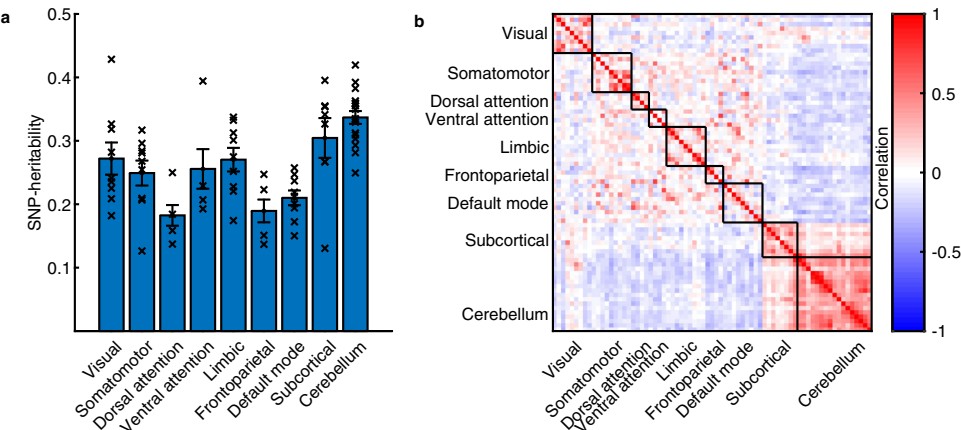

**Fig. 3 Mapping of SNP-based heritability and genetic correlation estimates obtained using MGREML ($N = 20{,}190$) of relative gray matter volumes in networks of intrinsic functional connectivity. a** Average SNP-based heritability (based on point estimates ×) of relative gray matter volume in networks of intrinsic functional connectivity (95% CI). **b** Genetic correlations in the brain in networks of intrinsic functional connectivity (blue vertices represent a negative correlation, red vertices a positive correlation).

volumes of the ROIs on the one hand and height and intelligence on the other hand (Supplementary Data 3). In the set of estimated correlations across the ROIs, the main differences with the results obtained using relative brain volumes (Supplementary Data 1) are that the genetic correlations within the cerebellum clusters are slightly smaller and that the positive correlations within the subcortical structures are somewhat larger.

## Discussion

We designed MGREML to estimate high-dimensional genetic correlation matrices from large-scale individual-level genetic data in a computationally efficient manner while guaranteeing the internal consistency of the estimated genetic correlation matrix. For comparison, we used pairwise bivariate GREML to obtain a genetic correlation matrix using the exact same set of individuals

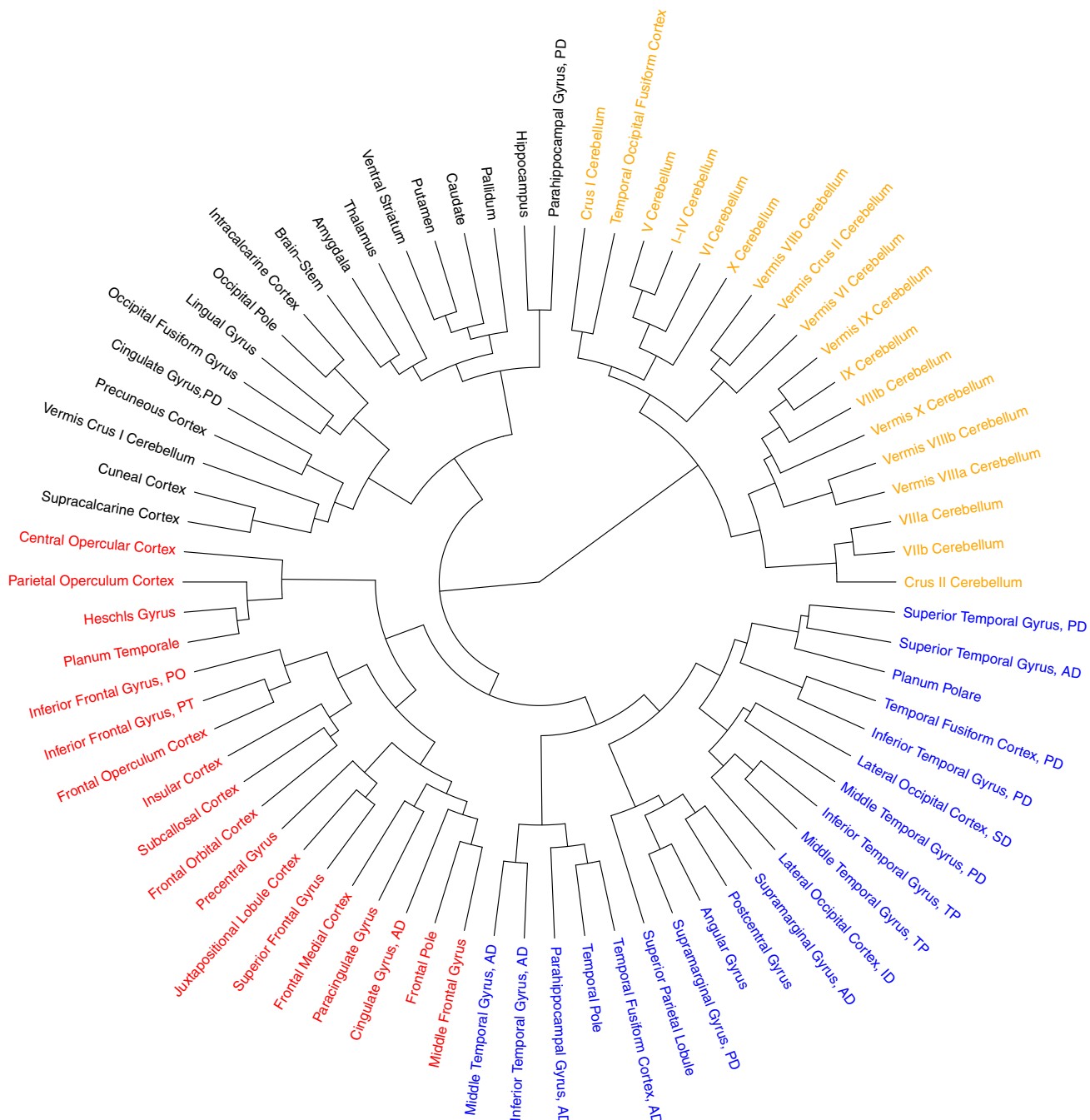

**Fig. 4 Dendogram of the estimated genetic correlations for the relative gray matter volumes of the 74 regions of interest in the brain.** Genetic correlations are estimated using MGREML ($N = 20{,}190$), and clusters are identified using Ward's method with a D2 ward for hierarchical clustering. Each color represents a different cluster.

($N = 20{,}190$) and traits ($T = 86$) as in our main analysis. While the resulting estimates are fairly similar (Supplementary Data 2), the resulting genetic correlation matrix is indefinite (13 out of the 86 eigenvalues are negative). Such an indefinite matrix poses a challenge for multivariate methods, such as Genomic SEM[42], that require a genetic correlation matrix as starting point for a follow-up analysis. Using MGREML results avoids this challenge, as MGREML by design guarantees the estimation of a positive (semi)-definite genetic correlation matrix.

Moreover, we conducted GWASs and bivariate LDSC[26] analyses to obtain a genetic correlation matrix using the pairwise bivariate approach for the same empirical application (Supplementary Data 5). We find that the standard errors of the $h^2_{\text{SNPs}}$

estimates obtained using MGREML are on average 32.7% smaller than those obtained using LDSC. The standard errors of the genetic correlations obtained using MGREML are on average 50.6% smaller compared to those obtained using LDSC, illustrating the advantages of MGREML in terms of statistical power. More specifically, when applying a two-sided significance test to each estimated genetic correlation (null hypothesis: $r_g = 0$; alternative hypothesis: $r_g \neq 0$), MGREML yields 1519 significant correlations at the 5% level, whereas the pairwise bivariate LDSC approach yields only 954 significant correlations. Thus, the gain in statistical efficiency is larger than the efficiency gained by HDL[34], a recently developed variation of bivariate LDSC that accounts for autocorrelation of summary statistics across the

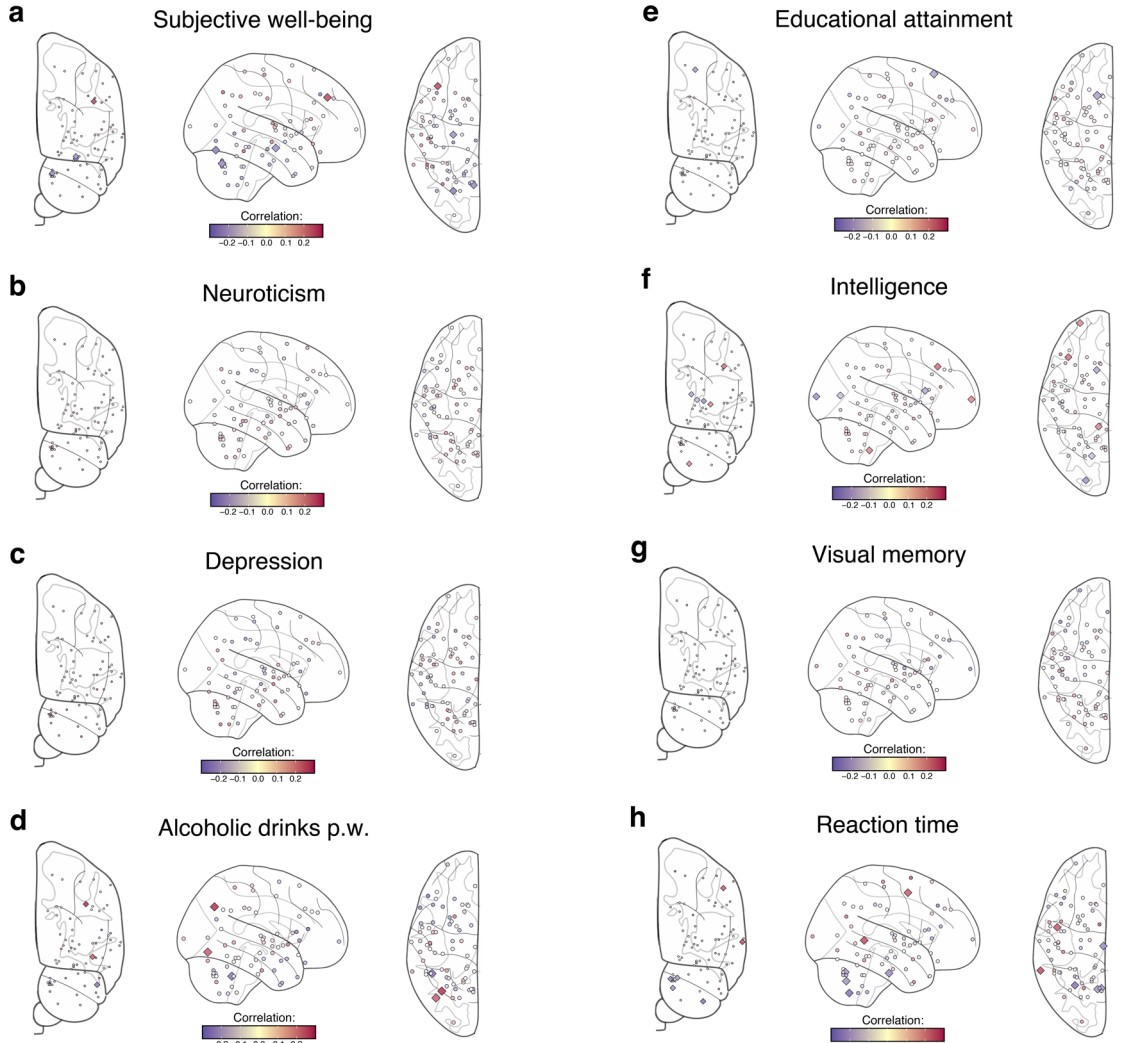

**Fig. 5 Spatial mapping of genetic correlation estimates obtained using MGREML. ($N = 20,190$) of relative gray matter volumes of the 74 regions of interest in the brain and 8 behavioral traits.** Blue and red points represent negative and positive genetic correlations, respectively. Diamonds represent estimates that are significant at the 5% level. **a** Subjective well-being. **b** Neuroticism. **c** Depression. **d** Alcoholic drinks per week. **e** Educational attainment. **f** Intelligence. **g** Visual spatial memory. **h** Reaction time.

genome as a result of LD. Importantly, the genetic correlation matrix obtained using bivariate LDSC is again not positive semidefinite and thus the estimated genetic correlations across traits are not internally consistent.

Our main results tacitly assume a homoscedastic per-SNP heritability, in line with GCTA[19]. This GCTA model approach may be suboptimal under some circumstances, including genetic drift and various forms of natural selection[52,53]. We therefore repeated the estimation of the genetic correlation matrix using the LDAK-Thin model[30,31] (Supplementary Data 6) and the SumHer[54] approach (Supplementary Data 7) that both assume heteroscedastic random SNP effects. Importantly, results based on the LDAK-Thin model can also be readily obtained using the MGREML software tool, because the choice of the heritability model only affects the construction of the genomic-relatedness matrix (GRM). Comparison of results shows that the heritability estimates are on average fairly similar across methods (Supplementary Data 8), and illustrates again that individual-level data methods (the GCTA model and LDAK-Thin model in MGREML) are statistically more efficient than summary statistics methods (LDSC and SumHer). In our empirical application, we find that the fit of MGREML in terms of the log-likelihood is slightly better when assuming the GCTA model than when assuming the LDAK-Thin model (Supplementary Note 3). The similarity of the estimates across different heritability models may be explained by differential selection across phenotypes, and balancing out of underestimations and overestimations of contributions to $h^2_{\text{SNPs}}$ in low- and high-LD regions[31,52].

Our results show marked variation in the estimated heritability across cortical gray matter volumes, with on average higher heritability estimates in subcortical and cerebellar areas than in cortical areas (Fig. 2b). Grouping of $h^2_{\text{SNPs}}$ estimates by networks of intrinsic functional connectivity suggests that heritability is particularly low in brain areas with presumed stronger experience-dependent plasticity (Fig. 3a). These results suggest that neocortical areas of the brain are under weaker genetic control perhaps reflecting greater environmentally determined plasticity[45,55]. Furthermore, the estimated genetic correlations suggest the presence of four genetically distinct clusters in the brain (Fig. 4). These clusters largely correspond with the conventional subdivision of the brain in different lobes based on anatomical borders[56]. The estimated genetic correlations also provide evidence for a shared genetic architecture of traits between which an association has been observed before in

phenotypic studies such as between intelligence and educational attainment[50]. In addition, genetic correlations were identified between alcohol consumption and cerebellar volume, and between subjective well-being and the temporooccipital part of the Middle Temporal Gyrus (Supplementary Data 1). We caution that these relationships may be somewhat different in the general population due to the nonrandom selection of the population into the UK Biobank sample[57] and potential gene–environment correlations[58].

To verify that our results are not merely a reflection of the physical proximity of brain regions, we regressed the estimated genetic correlations on the physical distance between the different brain regions. Although this correction procedure decreased the estimated genetic correlations by 17.4%, the main patterns are still observed. For the same reason, we recreated the dendogram (Fig. 3) after aggregating the results for subregions into an average for the larger region because the optimization procedure of MGREML puts equal weight on each trait and does not account for physical proximity. The results of this robustness check show that the four identified clusters do not merely reflect the number of analyzed measures for a specific brain region.

Estimates of heritability increase our understanding of the relative impact of genetic and environmental variation on traits[14,32], and estimates of genetic correlation lead to a better understanding of the shared biological pathways between traits[59]. Joint analysis of multiple traits may also improve the predictive power of genetic models[60]. MGREML has been designed to estimate both SNP-based heritability and genetic correlations in a computationally efficient and internally consistent manner using individual-level genetic data. The efficiency of its optimization algorithm makes it possible to use MGREML to estimate high-dimensional genetic correlation matrices in large datasets, such as the UK Biobank.

## Methods

**Sample and data**. Participants of this study were sourced from UK Biobank. UK Biobank is a prospective cohort study in the UK that collects physical, health, and cognitive measures, and biological samples (including genotype data) in about 500,000 individuals[8]. In 2016, UK Biobank started to collect brain imaging data with the aim to scan 100,000 subjects by 2022[27,61]. UK Biobank has received ethical approval from the National Health Service North West Centre for Research Ethics Committee (11/NW/0382) and has obtained informed consent from its participants.

We selected the 43,691 individuals with available genotype data from the UK Biobank brain imaging study who self-identified as 'white British' and with similar genetic ancestry based on a principal component analysis. After stringent quality control (Supplementary Note 4), we estimated pairwise genetic relationships using 1,384,830 autosomal common (Minor Allele Frequency ≥ 0.01) SNPs and retained 37,392 individuals whose pairwise relationship was estimated to be less than 0.025 (approximately corresponding to second- or third-degree cousins or more distant shared ancestry). From these unrelated individuals, we retained the 20,190 individuals (9747 males and 10,433 females) with complete information on all 86 traits in our analyses. The age of these individuals ranges from 40 to 72 years, and the average age is 54.79 years.

A description of all the variables used in the empirical analyses is available in Supplementary Note 2. Mapping of each cortical region to a network of intrinsic functional connectivity (Fig. 3) is based on the assignment of each brain parcel in the Harvard-Oxford atlas[62] to the intrinsic functional connectivity network[44] with the highest overlap. These networks were earlier identified using functional magnetic resonance imaging[44].

**Statistical framework**. In a genome-wide association study (GWAS) of quantitative trait $y$, the effect of single-nucleotide polymorphism (SNP) $m$ on $y$ is modelled as:

$$y_j = g_{jm}^* \alpha_m^* + \mathbf{x}_j' \boldsymbol{\beta} + u_j, \tag{1}$$

where $y_j$ is the phenotype of individual $j$ and $g_{jm}^*$ is the raw genotype (i.e., a value equal to zero, one, or two, indicating the number of copies of the coded allele) for the same individual and the given SNP. In this model, $\alpha_m^*$ is the per-allele effect of SNP $m$ on $y$, $\mathbf{x}_j'$ is a $1 \times k$ vector of control variables with $k \times 1$ vector of effects $\boldsymbol{\beta}$, and $u_j$ is the error term.

If $y$ has mean zero and/or an intercept is included in the set of control variables, we can assume, without loss of generality, that SNPs are standardized in accordance with their distribution under Hardy–Weinberg equilibrium. That is, we define $g_{jm} = (g_{jm}^* - 2f_m)[2f_m(1-f_m)]^{-0.5}$, where $g_{jm}$ denotes the standardized genotype for individual $j$ and SNP $m$, and where $f_m$ denotes the empirical allele frequency of the same SNP. Now, $g_{jm}^* \alpha_m^*$ in Eq. (1) can be replaced by $g_{jm} \alpha_m$, where $\alpha_m = \alpha_m^*[2f_m(1-f_m)]^{0.5}$ is the effect of standardized SNP $m$. In addition, we can consider the contribution of all SNPs jointly using the following model:

$$y_j = \mathbf{g}_j' \boldsymbol{\alpha} + \mathbf{x}_j' \boldsymbol{\beta} + \varepsilon_j, \text{ where } \mathbf{g}_j' \boldsymbol{\alpha} = g_{j1}\alpha_1 + \ldots + g_{jM}\alpha_M. \tag{2}$$

Here, $\mathbf{g}_j'$ is the $1 \times M$ vector of standardized genotypes for individual $j$, $\boldsymbol{\alpha}$ is the $M \times 1$ vector of effects, and $\varepsilon_j$ is the error term in this model. For a sample of $N$ individuals (Fig. 1, Panel a), Eq. (2) can be written in matrix notation as:

$$\mathbf{y} = \mathbf{G}\boldsymbol{\alpha} + \mathbf{X}\boldsymbol{\beta} + \boldsymbol{\varepsilon}, \tag{3}$$

where $\mathbf{G}$ is the $N \times M$ matrix of standardized genotypes, $\mathbf{X}$ is the $N \times k$ matrix of control variables, and $\boldsymbol{\varepsilon}$ is the $N \times 1$ vector of errors. In genomic-relatedness-based restricted maximum likelihood (GREML)[32] as implemented in GCTA[19], $\boldsymbol{\beta}$ is assumed to be fixed and SNP effects and errors are assumed to be random, $viz.$, $\boldsymbol{\alpha} \sim N(\mathbf{0}, \mathbf{I}_M \sigma_\alpha^2)$ and $\boldsymbol{\varepsilon} \sim N(\mathbf{0}, \mathbf{I}_N \sigma_E^2)$, where $\sigma_\alpha^2$ is the variance in SNP effects and $\sigma_E^2$ the variance in errors. Now, $\mathbf{G}\boldsymbol{\alpha}$ is the total genetic contribution, which follows a $N(\mathbf{0}, \mathbf{G}\mathbf{G}'\sigma_\alpha^2)$ distribution. Under this model, the phenotypic variance matrix across individuals can be decomposed as:

$$\text{Var}(\mathbf{y}) = \mathbf{A}\sigma_G^2 + \mathbf{I}_N \sigma_E^2, \tag{4}$$

where $\mathbf{A} = M^{-1}\mathbf{G}\mathbf{G}'$ is the genomic-relatedness matrix (GRM), capturing genetic similarity between individuals based on all SNPs under consideration (Fig. 1, Panel b), and $\sigma_G^2 = M\sigma_\alpha^2$ is the total contribution of additive, linear effects of SNPs to phenotypic variance. The SNP-based heritability $h_{\text{SNPs}}^2$ of $y$ is then defined as:

$$h_{\text{SNPs}}^2 = \frac{\sigma_G^2}{\sigma_G^2 + \sigma_E^2}. \tag{5}$$

Importantly, $\boldsymbol{\alpha} \sim N(\mathbf{0}, \mathbf{I}_M \sigma_\alpha^2)$ is equivalent to assuming all SNPs explain the same proportion of phenotypic variance. As a result, this assumption about SNP effects tacitly imposes a strong relation between allele frequencies and effect sizes, where the per-allele effects of rare variants are, on average, considerably larger than the per-allele effects of more common variants. Moreover, this assumption does not differentiate between regions of low and high linkage disequilibrium (LD). Therefore, other perhaps more realistic assumptions about the distribution of SNP effects have been proposed and utilized[30,31].

These alternatives typically only affect the way in which GRM $\mathbf{A}$ in Eq. (4) is constructed. More specifically, when heteroscedastic SNP effects (i.e., $\boldsymbol{\alpha} \sim N(\mathbf{0}, \mathbf{D}\sigma_\alpha^2)$) are assumed (with $\mathbf{D}$ a diagonal matrix reflecting, e.g., the strength of the relationship between allele frequencies and effect sizes), it follows that $\mathbf{G}\boldsymbol{\alpha} = \mathbf{G}\mathbf{D}^{0.5}\boldsymbol{\alpha}^*$, where $\boldsymbol{\alpha}^* \sim N(\mathbf{0}, \mathbf{I}_M \sigma_\alpha^2)$. In this case, by defining $\mathbf{A} = d^{-1}\mathbf{G}\mathbf{D}\mathbf{G}'$, with $d$ being the sum of the diagonal elements of $\mathbf{D}$, Eqs. (4) and (5) still apply. As such, our model also lends itself well for application to a GRM that is calculated using alternatives to GCTA[19], such as LDAK[31].

Irrespective of the precise definition of $\mathbf{A}$, we can write the model in Eq. (3) as:

$$\mathbf{y} \sim N(\mathbf{X}\boldsymbol{\beta}, \sigma_G^2\mathbf{A} + \sigma_E^2\mathbf{I}_N). \tag{6}$$

For two quantitative traits, observed in the same set of $N$ individuals, this model can be generalized to the following bivariate model[18]:

$$\begin{pmatrix} \mathbf{y}_1 \\ \mathbf{y}_2 \end{pmatrix} \sim N\left( \begin{pmatrix} \mathbf{X}_1 & \mathbf{0} \\ \mathbf{0} & \mathbf{X}_2 \end{pmatrix} \begin{pmatrix} \boldsymbol{\beta}_1 \\ \boldsymbol{\beta}_2 \end{pmatrix}, \begin{pmatrix} \sigma_{G_{11}}\mathbf{A} & \sigma_{G_{12}}\mathbf{A} \\ \sigma_{G_{12}}\mathbf{A} & \sigma_{G_{22}}\mathbf{A} \end{pmatrix} + \begin{pmatrix} \sigma_{E_{11}}\mathbf{I}_N & \sigma_{E_{12}}\mathbf{I}_N \\ \sigma_{E_{12}}\mathbf{I}_N & \sigma_{E_{22}}\mathbf{I}_N \end{pmatrix} \right), \tag{7}$$

where $\mathbf{X}_1$ (resp. $\mathbf{X}_2$) is the $N \times k_1$ ($N \times k_2$) matrix of control variables for trait $\mathbf{y}_1$ ($\mathbf{y}_2$) with fixed effects $\boldsymbol{\beta}_1$ ($\boldsymbol{\beta}_2$), $\sigma_{G_{st}}$ is the genetic covariance and $\sigma_{E_{st}}$ the environmental covariance between traits $s$ and $t$, for $s = 1, 2$ and $t = 1, 2$. The Kronecker product (denoted by '⊗') can be used to extend the model in Eq. (7) to a multivariate model for $T$ different traits (i.e., $\mathbf{y}_t$ for $t = 1, \ldots, T$), as follows[60,63]:

$$\begin{pmatrix} \mathbf{y}_1 \\ \mathbf{y}_2 \\ \vdots \\ \mathbf{y}_T \end{pmatrix} \sim N\left( \begin{pmatrix} \mathbf{X}_1 & \mathbf{0} & \mathbf{0} \\ \mathbf{0} & \ddots & \mathbf{0} \\ \mathbf{0} & \mathbf{0} & \mathbf{X}_T \end{pmatrix} \begin{pmatrix} \boldsymbol{\beta}_1 \\ \vdots \\ \boldsymbol{\beta}_T \end{pmatrix}, \mathbf{V}_G \otimes \mathbf{A} + \mathbf{V}_E \otimes \mathbf{I}_N \right), \tag{8}$$

where

$$\mathbf{V}_G = \begin{pmatrix} \sigma_{G_{11}} & \cdots & \sigma_{G_{1T}} \\ \vdots & \ddots & \vdots \\ \sigma_{G_{1T}} & \cdots & \sigma_{G_{TT}} \end{pmatrix} \text{ and } \mathbf{V}_E = \begin{pmatrix} \sigma_{E_{11}} & \cdots & \sigma_{E_{1T}} \\ \vdots & \ddots & \vdots \\ \sigma_{E_{1T}} & \cdots & \sigma_{E_{TT}} \end{pmatrix}. \tag{9}$$

In this multivariate model, the SNP-based heritability ($h_{\text{SNPs}}^2$) of trait $t$, denoted by $h_{\text{SNPs}}^2(t)$, and the genetic correlation ($r_g$) between traits $s$ and $t$ (Fig. 1, Panel c),

denoted by $r_g(s, t)$, are defined as:

$$h^2_{\text{SNPs}}(t) = \frac{\sigma_{G_{tt}}}{\sigma_{G_{tt}} + \sigma_{E_{tt}}} \text{ and } r_g(s, t) = \frac{\sigma_{G_{st}}}{\sqrt{\sigma_{G_{tt}}\sigma_{G_{ss}}}}, \tag{10}$$

for $s = 1, \ldots, T$ and $t = 1, \ldots, T$.

**Optimization procedure.** To estimate the genetic and environmental covariance matrices $\mathbf{V}_G$ and $\mathbf{V}_E$ in Eqs. (8) and (9), we use restricted maximum likelihood (REML) estimation. To maximize the likelihood function, we use a quasi-Newton method. More specifically, we use a Broyden–Fletcher–Goldfarb–Shanno (BFGS) algorithm[64]. Supplementary Note 1 provides highly efficient expressions for the log-likelihood and gradient, which are needed in the optimization algorithm. These expressions make it possible to estimate the multivariate model with a time complexity that scales linearly with the number of observations and quadratically with the number of traits. The optimization procedure guarantees that the estimated matrices $\mathbf{V}_G$ and $\mathbf{V}_E$ are positive (semi)-definite, by imposing an underlying factor model for both matrices. After optimization, standard errors can be calculated with a time complexity that scales linearly with the number of observations and quadratically with the number of parameters in the model (which in turn scales quadratically with the number of traits). This optimization procedure is fully incorporated in MGREML, a command-line tool written in Python 3. We recommend using the GCTA-GREML power calculator[65] for ex-ante power calculations, because the accuracy of estimates from MGREML and pairwise bivariate GREML is fairly similar (Supplementary Data 8).

**Statistics and reproducibility.** The empirical results in this study have been obtained using the command-line tool MGREML. Supplementary Note 4 details the analysis pipeline that has been used to obtain the heritability and genetic correlation estimates.

**Reporting summary.** Further information on research design is available in the Nature Research Reporting Summary linked to this article.

## Data availability

Individual-level genotype and phenotype data are available by application via the UKB Biobank website (https://www.ukbiobank.ac.uk/). The authors declare that the results supporting the findings of this study are available within the paper and its supplementary files. Figures 2–5 are based on the MGREML results available in Supplementary Data 1.

## Code availability

MGREML is available at https://github.com/devlaming/mgreml as a ready-to-use command-line tool[66]. The GitHub page comes with a full tutorial on the usage of this tool. An MGREML analysis of 86 traits, observed in a sample of 20,190 unrelated individuals (i.e., the dimensionality of the dataset that we use in our empirical application), takes around four hours on a four-core laptop with 16GB of RAM.

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

## Acknowledgements

UK Biobank has obtained ethical approval from the National Research Ethics Committee (11/NW/0382). This research has been conducted using the UK Biobank Resource under application number 11425. We would like to thank the participants and researchers from UK Biobank Imaging Study who contributed or collected data. We also thank the Pan-UKB team for providing the UK Biobank specific LD scores (https://pan.ukbb.broadinstitute.org). This work was carried out on the Dutch national e-infrastructure with the support of SURF Cooperative (NWO Call for Compute Time EINF-403 to E.A.W.S.). P.D.K. and R.d.V. were supported by a European Research Council Consolidator Grant (647648 EdGe to P.D.K.). P.D.K. was also supported by the Office of the Vice Chancellor for Research and Graduate Education at the University of Wisconsin–Madison with funding from the Wisconsin Alumni Research Foundation. C.A.R. was supported by a European Research Council Starting Grant (946647 GEPSI). The funders had no role in study design, data collection and analysis, decision to publish or preparation of the manuscript.

## Author contributions

R.d.V., E.A.W.S., and P.J.F.G. developed the model. R.d.V., E.A.W.S., P.D.K., and C.A.R. designed the experiments. R.d.V. and E.A.W.S. wrote code and performed the statistical analyses. R.d.V., E.A.W.S., P.R.J., A.D., P.D.K., and C.A.R. analyzed the results. E.A.W.S. and P.R.J. visualized the results. C.A.R. led the preparation of the manuscript and supplementary files. All authors contributed to the editing of the manuscript and supplementary files.

## Competing interests

The authors declare no competing interests.
