## [Peer Review File · Communications Biology]

REVIEWERS' COMMENTS:

Reviewer #1 (Remarks to the Author):

The authors have provided very detailed responses to my comments. I am particularly grateful for them taking the time to answer so fully my "curiosity questions" - the answers were very interesting to read. It also seems they have carefully answered the comments of the other reviewers.

Nonetheless, I have two outstanding points, one major, one minor.

1 (Major) - I really appreciate you repeating the analysis switching the GCTA Model with the LDAK-Thin Model, thank you. I also appreciate that you have added a detailed paragraph in the discussion explaining the two choices of heritability model. However, I am not entirely happy with your justification for continuing to keep the GCTA Model as the main results. You say the following:

"Given the high similarity of the results, we decided to keep the MGREML estimates using the "GCTA Model" as main results, as most GCTA users will be intimately familiar with this model, despite the advantages of LDAK. We are happy to reconsider this choice upon your guidance as well as the editor's."

My view is your main results should be the ones you think are most accurate. Your method produces a likelihood - ie a measure of model fit, which indicates how well the data fit each model. Therefore, I would recommend reporting results from the GCTA Model if this leads to highest likelihood, and reporting results from the LDAK-Thin Model if this leads to highest likelihood.

I understand your concern that deviating from the GCTA Model will make the paper harder to understand. I agree that it is more complicated to describe the general case, instead of that corresponding to the GCTA Model. However, hopefully you could minimize the added confusion, and this would be compensated for by the scientific value. It seems you would not have to change anything in the main text (as this provides only a summary of the mathematical concepts, that do not depend on the choice of heritability model). Instead, you would only have to add a few lines to the (online) Methods.

For example, when you currently say in Line 324
"where G is the $N \times M$ matrix of standardized genotypes"
you could instead say

"where G is the $N \times M$ matrix of genotypes standardized to have mean zero and variance q_j , where the constants q_j are determined by the choice of heritability model (see below)"

By incorporating the heritability model within G, there is then no need to change any subsequent maths (except possibly replace XXT/M by XXT/Q , where Q is the sum of q_j).

If the LDAK-Thin Model fits better than the GCTA Model, you could then add:

"The heritability model specifies the expected heritability contributed by each SNP. It is common to assume each SNP is expected to contribute equally and set $q_j=1$, which is referred to as the GCTA Model. For our main analyses, we instead used the LDAK-Thin Model and set $q_j=I_j [p_j(1-p_j)]^{0.75}$, where I_j indicates whether SNP j remains after thinning for duplicates [I believe you have already defined p_j by here], as this resulted in a higher likelihood than the GCTA Model, indicating the former better reflects the genetic architecture of the brain traits."

On the other hand, if the GCTA Model fits better than the LDAK-Thin Model, you could add:

"The heritability model specifies the expected heritability contributed by each SNP. For our main analysis, we set $q_j=1$, which assumes that each SNP is expected to contribute equally (referred to as the GCTA Model). For a secondary analysis, we instead set q_j according to the LDAK-Thin Model. However, this resulted in a lower likelihood than the GCTA Model, indicating the GCTA Model better reflects the genetic architecture of the brain traits."

Note that if you did make these changes, then I think you would be justified reducing the section about heritability models in the discussion (if you wished).

Again, I appreciate you taking seriously my comment about the heritability model. I have stated my opinion, but I am happy for the editor to offer their opinion, or alternatively, if they decide to ask the other two reviewers for their views.

2 (Minor) - There was small confusion about my LDSC comment (number 9). When I said "same data", I meant computing LDscores yourself using the 20k UKBB individuals and the same SNPs used with MTGREML. Instead, I think the LDscores computed by the PanUKB team used a more dense set of SNPs for a larger number of UKBB individuals. My comment that your original analysis was "not quite fair to LDSC" was primarily in reference to the fact they use more dense SNPs than you, and thus you would expect their genetic correlations to be different (and possibly less precise, it is hard for me to say). However, this was already a very minor point, so I don't want to burden you with extra analyses.

Signed Doug Speed

Reviewer #2 (Remarks to the Author):

All of my concerns were adequately addressed.

Reviewer #3 (Remarks to the Author):

The authors have addressed most of my concerns and the manuscript has been significantly improved. I have no further comments.

Reviewer 1

The authors have provided very detailed responses to my comments. I am particularly grateful for them taking the time to answer so fully my “curiosity questions” - the answers were very interesting to read. It also seems they have carefully answered the comments of the other reviewers. Nonetheless, I have two outstanding points, one major, one minor.

I (Major) - I really appreciate you repeating the analysis switching the GCTA Model with the LDAK-Thin Model, thank you. I also appreciate that you have added a detailed paragraph in the discussion explaining the two choices of heritability model. However, I am not entirely happy with your justification for continuing to keep the GCTA Model as the main results. You say the following:

“Given the high similarity of the results, we decided to keep the MGREML estimates using the “GCTA Model” as main results, as most GCTA users will be intimately familiar with this model, despite the advantages of LDAK. We are happy to reconsider this choice upon your guidance as well as the editor’s.”

My view is your main results should be the ones you think are most accurate. Your method produces a likelihood - ie a measure of model fit, which indicates how well the data fit each model. Therefore, I would recommend reporting results from the GCTA Model if this leads to highest likelihood, and reporting results from the LDAK-Thin Model if this leads to highest likelihood.

I understand your concern that deviating from the GCTA Model will make the paper harder to understand. I agree that it is more complicated to describe the general case, instead of that corresponding to the GCTA Model. However, hopefully you could minimize the added confusion, and this would be compensated for by the scientific value. It seems you would not have to change anything in the main text (as this provides only a summary of the mathematical concepts, that do not depend on the choice of heritability model). Instead, you would only have to add a few lines to the (online) Methods.

For example, when you currently say in Line 324

“where G is the $N \times M$ matrix of standardized genotypes”

you could instead say

“where G is the $N \times M$ matrix of genotypes standardized to have mean zero and variance q_j , where the constants q_j are determined by the choice of heritability model (see below)”

By incorporating the heritability model within G , there is then no need to change any subsequent maths (except possibly replace XXT/M by XXT/Q , where Q is the sum of q_j).

If the LDAK-Thin Model fits better than the GCTA Model, you could then add:

“The heritability model specifies the expected heritability contributed by each SNP. It is common to assume each SNP is expected to contribute equally and set $q_j=1$, which is referred to as the GCTA Model. For our main analyses, we instead used the LDAK-Thin Model and set $q_j=I_j [p_j(1-p_j)]^{0.75}$, where I_j indicates whether SNP j remains after thinning for duplicates [I believe you have already defined p_j by here], as this resulted in a higher likelihood than the GCTA Model, indicating the former better reflects the genetic architecture of the brain traits.”

On the other hand, if the GCTA Model fits better than the LDAK-Thin Model, you could add:

“The heritability model specifies the expected heritability contributed by each SNP. For our main analysis, we set $q_j=1$, which assumes that each SNP is expected to contribute equally (referred to as the GCTA Model). For a secondary analysis, we instead set q_j according to the LDAK-Thin Model. However, this resulted in a lower likelihood than the GCTA Model, indicating the GCTA Model better reflects the genetic architecture of the brain traits.”

Note that if you did make these changes, then I think you would be justified reducing the section about heritability models in the discussion (if you wished).

Again, I appreciate you taking seriously my comment about the heritability model. I have stated my opinion, but I am happy for the editor to offer their opinion, or alternatively, if they decide to ask the other two reviewers for their views.

We thank the reviewer for raising these further considerations regarding the choice of the most appropriate heritability model. A quick inspection of the log-likelihoods of the MGREML analyses using the ‘GCTA Model’ and the ‘LDAK Thin Model’ reveals that they yield virtually identical fit in our empirical application, with the log-likelihood of the ‘GCTA Model’ being marginally higher than that of the ‘LDAK Thin Model’:

- 1) Log-likelihood under the ‘GCTA Model’ = -2147476.79,
- 2) Log-likelihood under the ‘LDAK Thin Model’ = -2147685.36.

This result strengthens our view that in this particular empirical application, estimates from the ‘GCTA Model’ should be most prominently featured in our manuscript.

However, cognizant of the fact that the choice of heritability model typically depends on the set of phenotypes considered in a given empirical application, we have added the following additional paragraph to the *Statistical framework* subsection of the **Methods** section, further clarifying how and why our method can easily incorporate different heritability models:

“Importantly, $\alpha \sim N(\mathbf{0}, \mathbf{I}_M \sigma_\alpha^2)$ is equivalent to assuming all SNPs explain the same proportion of phenotypic variance. As a result, this assumption about SNP effects tacitly imposes a strong relation between allele frequencies and effect sizes, where the per-allele effects of rare variants are, on average, considerably larger than the per-allele effects of more common variants. Moreover, this assumption does not differentiate between regions of low and high linkage disequilibrium (LD). Therefore, other perhaps more realistic assumptions about the distribution of SNP effects have been proposed and utilized^{Error! Reference source not found., Error! Reference source not found.}.

These alternatives typically only affect the way in which GRM \mathbf{A} in Equation 4 is constructed. More specifically, when heteroscedastic SNP effects (i.e., $\alpha \sim N(\mathbf{0}, \mathbf{D} \sigma_\alpha^2)$) are assumed (with \mathbf{D} a diagonal matrix reflecting, e.g., the strength of the relationship between allele frequencies and effect sizes), it follows that $\mathbf{G}\alpha = \mathbf{G}\mathbf{D}^{0.5}\alpha^*$, where $\alpha^* \sim N(\mathbf{0}, \mathbf{I}_M \sigma_\alpha^2)$. In this case, by defining $\mathbf{A} = d^{-1}\mathbf{G}\mathbf{D}\mathbf{G}'$, with d being the sum of the diagonal elements of \mathbf{D} , Equations 4 and 5 still apply. As such, our model also lends itself well for application to a GRM that is calculated using alternatives to GCTA^{Error! Reference source not found.}, such as LDAK^{Error! Reference source not found.}.

2 (Minor) - There was small confusion about my LDSC comment (number 9). When I said “same data”, I meant computing LD Scores yourself using the 20k UKBB individuals and the same SNPs used with MTGREML. Instead, I think the LD Scores computed by the PanUKB team used a more dense set of SNPs for a larger number of UKBB individuals. My comment that your original analysis was “not quite fair to LDSC” was primarily in reference to the fact they use more dense SNPs than you, and thus you would expect their genetic correlations to be different (and possibly less precise, it is hard for me to say). However, this was already a very minor point, so I don't want to burden you with extra analyses.

Point taken. We note that the vast majority of LDSC users will make use of precomputed LD scores. As such, we believe the current results to be most valuable from a practical perspective.

Reviewer 2

All of my concerns were adequately addressed.

Reviewer 3

The authors have addressed most of my concerns and the manuscript has been significantly improved. I have no further comments.